# Sudden Intrauterine Unexplained Death (SIUD) and Oxidative Stress: Placental Immunohistochemical Markers

**DOI:** 10.3390/cells13161347

**Published:** 2024-08-13

**Authors:** Angelo Montana, Letizia Alfieri, Raffaella Marino, Pantaleo Greco, Cristina Taliento, Ezio Fulcheri, Anastasio Tini, Francesca Buffelli, Margherita Neri

**Affiliations:** 1Department of Biomedical Sciences and Public Health, University Politecnica delle Marche, 60126 Ancona, Italy; anastasio.tini78@gmail.com; 2Department of Medical Sciences, Section of Legal Medicine, University of Ferrara, Via Fossato di Mortara 70, 44121 Ferrara, Italy; raffaella.marino1@gmail.com (R.M.); margherita.neri@unife.it (M.N.); 3Department of Medical Sciences, Section of Obstetrics and Gynecology, University of Ferrara, Via Aldo Moro 8, 44124 Ferrara, Italy; pantaleo.greco@unifg.it (P.G.); cristina.taliento1@gmail.com (C.T.); 4Division of Anatomic Pathology, Department of Surgical and Diagnostic Sciences (DISC), University of Genova, 16148 Genoa, Italy; ezio.fulcheri@unige.it; 5Fetal-Perinatal Pathology Unit, IRCCS-Istituto Giannina Gaslini, 16147 Genoa, Italy; francescabuffelli@gaslini.org

**Keywords:** SIUD, placenta, immunohistochemical markers, forensic pathology, oxidative stress

## Abstract

Background: Intrauterine fetal death and perinatal death represent one of the most relevant medical scientific problems since, in many cases, even after extensive investigation, the causes remain unknown. The considerable increase in medical legal litigation in the obstetrical field that has witnessed in recent years, especially in cases of stillborn births, has simultaneously involved the figure of the forensic pathologist in scientific research aimed at clarifying the pathophysiological processes underlying stillbirth. Methods: our study aims to analyze cases of sudden intrauterine unexplained death syndrome (SIUD) to evaluate the role of oxidative stress in the complex pathogenetic process of stillbirth. In particular, the immunohistochemical expression of specific oxidative stress markers (NOX2, NT, iNOS, 8-HODG, IL-6) was evaluated in tissue samples of placentas of SIUDs belonging to the extensive case series (20 cases), collected from autopsy cases of the University of Ferrara and Politecnica delle Marche between 2017 and 2023. Results: The study demonstrated the involvement of oxidative stress in intrauterine fetal deaths in the placenta of the cases examined. In SIUD, the most expressed oxidative stress markers were NOX2 and 8-HODG. Conclusions: The study contributes to investigating the role of oxidative stress in modulating different pathways in unexplained intrauterine fetal death (SIUD) tissues.

## 1. Introduction

SIUD is a cluster of fetal deaths (≥25 gestational weeks) where both the maternal history and complete (i.e., inclusive of autopsy, histology of fetus and placenta, microbiology, etc.) post-mortem examination have ruled out any recognized cause [1,2].

Matturri et al. defines SIUD as “The sudden death during pregnancy that remains unexplained after an in-depth autopsy including examination of the placental disk, umbilical cord and membranes, detailed pregnancy history analysis and molecular and microbiological investigations”. Sudden intrauterine unexplained/unexpected death syndrome, or unexpected stillbirth, is defined as “The late fetal death before the complete expulsion or removal of the fetus from the mother ≥ 25 weeks of gestation which is unexpected by history and is unexplained after review of the maternal clinical history and the performance of a general autopsy of the fetus, including examination of the placental disk, umbilical cord and membranes, and microbiological and genetic investigations” [1,3].

Unexplained stillbirths represent about 50% of perinatal deaths, with a prevalence of 5–12 cases per 1000 [4], and, in Italy, the incidence of SIUD, according to the forensic application of Italian Law (Number. 31/2006. About “Regulations for diagnostic post-mortem investigation in victims of sudden infant death syndrome (SIDS) and unexpected fetal death”) is estimated to be around 4–5% of births occurring after 25 weeks of pregnancy [1].

In Italy, the estimated perinatal death rate was 4.1 out of 1000 births in the last ten years, with variability by geographical area, which is worthy of further study, which sees the highest rate in Sicily (4.7/1000) and the lowest values in Lombardy (3.6/1000) and Tuscany (2.7/1000) [5].

Maternal age, low education, fetal male sex, and maternal high cigarette consumption are risk factors for SIUD. Another interesting finding is that about 50% of SIUD cases were considered to be cases of intrauterine growth restriction (IUGR) [6].

Oxidative stress (OS), therefore, derives from an abnormal increase in reactive oxygen species (ROS), primarily produced during oxygen reduction reactions. When these molecules exceed their average physiological value, they cannot be neutralized by the antioxidant defense systems. Thus, oxidative stress results from an imbalance between ROS production and antioxidant defense, leading to cellular damage and potential apoptosis [7,8].

The factors responsible for oxidative stress include the following: reactive nitrogen species (RNS), hydroxyl radical, superoxide anion, hydrogen peroxide, antioxidant systems, superoxide dismutase (SOD), catalase, carotenoids, glutathione peroxidase, and thiols [9,10,11,12,13,14,15,16].

Identifying oxidative stress markers has been the focus of many studies, and several markers have been proposed over the past decades. However, for some of them, there is a lack of consensus concerning standardization, reproducibility, and validation [17,18].

ROS have an essential function in regulating circulatory physiology, but can also actively contribute to the development of severe placenta-related obstetrical complications, such as intrauterine growth restriction and pre-eclampsia [9,10,11,12]. In the first trimester, OS stimulates the development of placental blood vessels and induces the expression of vascular endothelial growth factor (VEGF) [9,10], while in the second and third trimesters, OS can lead to massive apoptosis and placental insufficiency through the overexpression of apoptotic regulators at the placental level [11,12].

The changes in placental function and pathology carry a twofold significance. Firstly, they serve as an indicator of chronic stress that may develop in all placentas as they approach term, stemming from a growing disparity between the maternal nutrients and oxygen supply and the fetal demands [19,20]. BOLD MRI has definitively shown transient reductions in placental oxygenation during subclinical uterine contractions [21], which become more exaggerated as contractions build toward delivery. Fluctuations in placental oxygenation represent a significantly more impactful OS stimulus than hypoxia alone [22]. The function of organelles may decline due to accumulated oxidative damage, potentially compromising placental function and increasing the risk of stillbirth [23].

After analyzing the problems relating to SIUD and oxidative stress, our study tested the placenta samples selected in our autopsy cases to identify immunohistochemical markers useful in the forensic field. A previous study evaluated OS in the timing of hypoxic–ischemic brain damage in newborns, evaluating the forensic applications of these markers [24]. In the current study, we aimed to examine oxidative stress markers in SIUD cases because these cases often lead to significant forensic malpractice investigations. We selected the following biomarkers as they showed greater validity and reproducibility for our study: NADPH Oxidase 2 (NOX2), 8-Hydroxy-2′-deoxyGuanosine (8OHdG), nitrotyrosine (NT), Inducible Nitric Oxide Synthase (iNOS), and Interleukin-6 (IL-6).

The study demonstrated the involvement of oxidative stress in intrauterine fetal deaths in the placenta of the cases examined. In SIUD, the most expressed oxidative stress markers were NOX 2 and 8-HODG.

## 2. Materials and Methods

### 2.1. Subjects and Data Collection

This retrospective study examined cases of SIUD. Placental samples were collected from two centers (the Department of Legal Medicine of the University of Ferrara and the Università Politecnica delle Marche) from January 2016 to December 2023.

We collected 15 autoptic SIUD cases (Group 1) selected from the 2 centers’ general database of autopsy cases. All SIUD cases were subjected to a complete post-mortem investigation.

From the medical documentation for each case, we collected comprehensive information about pregnancy, fetal development, and delivery. The data regarding where the death happened, such as environmental and familiar situations and potential risk factors like, for example, maternal smoking or maternal obesity, were gathered. 

For each case, we reviewed both the autopsy report and the microscopic histopathological analysis focusing on tissue collected during autopsies after being stained with hematoxylin-eosin. In all cases, maternal diseases, fetal malformations, and placental and cord pathologies were absent. We selected cases of SIUD where gestational age ranged between 37 and 41 weeks and where no anomaly was detectable both by histology and medical history. Maternal age was, on average, 35 (span 19–45). Thirteen cases were delivered vaginally, while twelve were delivered through cesarean section. Having discarded all cases of fetal restriction (Intra Uterine Growth Restriction/Fetal Growth Restriction: FGR/IUGR), birth weight was within the normal range.

In order to test the value of our results, we introduced a control group (Group 2) of women who delivered in the same period, in the same institutions, without recorded abnormalities in the medical records and full placental examination. We carried out the same experimental work-out on their placentas.

A summary of cases selected for the two groups is summarized in Table 1.

### 2.2. Methods

All placental samples were evaluated by immunohistochemical staining using antibodies to NOX2, 8OHdG, NT, iNOS, and IL-6. For each case, sections with a thickness of 4 μm were created using a microtome. The sections, placed on a special slide, were hydrated and then subjected to pre-treatment for antigen recovery. Each tissue section was incubated with primary antibodies; the dilution are specified in Table 2. A refined avidin-biotin detection system was used in which the biotinylated secondary antibody reacts with several peroxidation-conjugated streptavidin molecules (CTS005 HRP-DAB system R&D kit, R&D Systems, Inc., Minneapolis, MN, USA). The positivity of the reaction was visualized by peroxidation of 3,3′-diaminobenzidine (DAB). The method described has already been validated and applied, as described in a previously published article [24].

#### Description of Oxidative Stress Markers

The summary of the markers analyzed with immunohistochemical reactions is clearly presented in Table 3.

**Table 2 cells-13-01347-t002:** Antibody dilution.

Primary Antibody	Dilution
NOX2 (Santa Cruz, CA, USA)	1:50
iNOS (Santa Cruz, CA, USA)	1:100
NT (Santa Cruz, CA, USA)	1:600
8-OHdG (JalCA, Tokyo, Japan)	1:10
IL-6 (Santa Cruz, CA, USA)	1:200

**Table 3 cells-13-01347-t003:** Markers of oxidative stress selected for the study.

Marker	Characteristics
NOX 2	Protein encoded by the CYBB gene in humans actively generates reactive oxygen species (ROS).
iNOS	The NOS2 gene in humans encodes the inducible form of enzyme, nitric oxide synthase.
NT	NT is a crucial marker of cell damage, inflammation, and NO production, formed as a result of tyrosine nitration mediated by reactive nitrogen species like peroxynitrite anion and nitrogen dioxide.
8-HODG	8-hydroxy-2′-deoxyguanosine is an oxidized derivate of deoxyguanosine and one of the significant products of DNA oxidation. Concentrations of 8-HODG serve as indicators of oxidative stress levels within the cell.
IL-6	IL-6 is a multifunctional cytokine secreted by macrophages and T lymphocytes to stimulate the immune response in various situations, such as trauma or infection. It can penetrate the blood–brain barrier.

Authors examined the slices using a Nikon Eclipse E600 microscope (Nikon, Tokyo, Japan). As previously illustrated, the ImageJ software 1.54g Java 1.8.0_345 (64 bit) (https://imagej.nih.gov/ij/) accessed on 5 April 2024 was used to quantify NOX2-, 8OHdG-, NT, iNOS and IL-6-positive-stained cells, [25,26], using the “Manual Cell Counting and Marking” protocol of this software for RGB color, single, not stacked images (https://imagej.nih.gov/ij/docs/guide/user-guide.pdf, accessed on 5 April 2024). Quantification was performed by counting the number of cells positively labeled by staining per analyzed area. The researchers who carried out the histological analyses were blinded to the information of the cases, and the data were blinded until the study was terminated.

### 2.3. Statistical Analysis

The statistical analysis was performed using the GraphPad Prism 10.2.3 software for Windows (La Jolla, CA, USA). The data were normalized and analyzed by choosing the unpaired *t*-test. The unpaired *t*-test allows the comparison of the difference between the means and the standard error of the difference, calculated by combining the standard errors of the two groups. A *p* value < 0.05 was considered statistically significant.

## 3. Results

### 3.1. NOX2

Immunohistochemical results of Group 1 (SIUD cases): anti-NOX2 antibodies showed a very intense and widespread positive reaction in placenta samples; all the control preparations show a basal expression of NOX2 (Figure 1).

### 3.2. NT

The placenta samples of Group 1 (SIUD cases) incubated with the anti-NT antibody revealed an intense and widespread positivity; the controls showed a basal positivity (Figure 2).

### 3.3. iNOS

Group 1 cases that were incubated with anti-iNOS antibodies revealed an intermediate widespread positivity. The control case samples also showed very mild immunopositivity in 100% of the slides (Figure 3).

### 3.4. 8-HODG

The immunohistochemical reaction of the Group 1 cases incubated with anti-8-HODG antibodies revealed a strong and widespread immunopositivity in all cases. Furthermore, 100% of the preparations relating to the control group showed basal immunopositivity (Figure 4).

### 3.5. IL-6

Cases of Group 1 incubated with anti-IL-6 antibodies revealed widespread positivity in all cases. The controls showed basal or midline positivity in the preparations (Figure 5).

The study, carried out on 15 cases subjected to the judicial autopsy of as many fetuses who died from SIUD, allowed the expression of oxidative stress to be evaluated on tissue samples using the immunohistochemistry method. All cases were tested for NOX 2, NT, iNOS, 8-HODG, and IL-6 in the placenta. All five markers used were positive and showed statistically significant expression compared to controls. It should be underlined that NOX2, 8-HODG, and NT showed a much more intense positivity than iNOS and IL-6, which was more significant. Figure 6 shows the graphical representation of the comparison of the expression among the various markers.

## 4. Discussion

OS is an essential factor in the pathological development of complicated pregnancies [27]. The literature suggests a link between oxidative stress and placental aging, which can have negative effects on pregnancy outcomes. Changes in cellular metabolism, such as telomere shortening and the dysfunction of telomerase, can induce an increase in oxidative metabolism, causing premature placental aging. OS can also induce the proliferation of pro-apoptotic inflammatory mediators. Placental damage occurs predominantly against DNA, lipids, and proteins, causing placental dysfunction and insufficiency. The endothelial damage induced by ROS and the decrease in antioxidant systems contribute to the formation of many pathologies complicating pregnancy, such as pre-eclampsia, IUGR, preterm births, and repeated abortion [28].

Oxidative stress is one of the pathophysiological mechanisms underlying preterm births (before 37 weeks of gestation). It has been suggested that cellular apoptosis may transmit an inflammatory signal that initiates birth. It is known that oxidative damage can compromise DNA and cause telomere shortening, thus accelerating the senescence of fetal membranes and causing inflammatory activation that contributes to delivery. Indeed, several studies observed an increase in the serum levels of placental and maternal oxidized metabolites (malondialdehyde, GSH) in preterm births.

Although the exact mechanisms behind SIUD are not fully understood, there have been frequent reports of developmental abnormalities, especially in the nervous and cardiac systems. Various abnormalities have been observed in the cardiac conduction system, such as the underdevelopment of central fibrous body cells, accessory atrioventricular pathways, and cartilaginous hyperplasia. Within the central nervous system, the arcuate nucleus (ARC) is a notable cardio-respiratory center located in the ventral medulla; in over 50% of SIUD cases, an ARC hypoplasia has been detected. Pulmonary immaturity has also been found in SIUD [4].

Among the many mechanisms that may be involved are X-linked congenital diseases, preterm labor, and poor fetal growth (i.e., intra-uterine growth restriction). The latter two disorders might be related to a higher incidence of placental vascular diseases, including pre-eclampsia.

Lattuada et al. found that OS may also play a role in the pathophysiology of SIUD. The above modifications may induce adaptive metabolic processes, such as the stimulation of mitochondrial biogenesis. Apart from producing ATP, mitochondria play a crucial role in cell metabolism, apoptosis, the generation of free radicals, and calcium balance. MtDNA mutations have been known to be involved in many diseases and syndromes. Therefore, ROS cause multiple mutations in the mitochondrial genome, leading to a reduction in the efficiency of mtDNA repair systems [29].

Stillbirths are often the result of a complex causal network where an already compromised fetus is more susceptible to infection and hypoxic insults [30].

Despite all the improvements made in maternal and infant care, even today, most (40–80%) of these deaths remain unexplained [8]. In a review published by De Bernis et al., it has emerged that, in our country, half of the late-born deaths (i.e., antepartum stillbirth and intrapartum; ≥28 weeks) also remain unexplained [31,32].

The results of our study about the effect of oxidative stress on placental tissue in SIUD cases, are very interesting; we demonstrated that oxidative stress is greatly improved in SIUD cases. NOX2, 8HOGD, and nitrotyrosine play essential roles in placental oxidative damage. We observed an increase in iNOS and IL-6 compared to the control group, with a statistically significant difference. However, the increase in these two markers is lower than that of NOX, 8HODG, and NT. 

According to the literature, these data are useful because when maternal blood reaches the placental level, there is an increase in oxygen tension, which is correlated with an increase in OS. Maternal blood commonly invades the placenta only after 11–12 weeks of gestation. The early presence of maternal blood before 10–11 weeks has been found to cause oxidative stress, leading to the deterioration of the syncytiotrophoblast and subsequently resulting in pregnancy loss. Certain indicators of placental oxidative damage are demonstrated in laboratory settings when placental villi are exposed to 21% oxygen, correlating with an escalation in local ROS production, and are expressed in vitro by exposing the placental villi to 21%, which is also associated with the increase in local ROS production [33].

According to some studies, there may be a connection between stillbirth and issues with the placenta such as infarction, thickening, calcification, and the dysfunction of placental vessels. In 2016, Ferrari et al. found that there was a significant decrease in the length of telomeres in the placentas associated with SIUD. This suggests that the placenta may undergo premature aging due to telomere-dependent senescence, which could lead to placental dysfunction and ultimately fetal death. It has finally been hypothesized that the change in placental proteins, lipids, and DNA induced by oxidative stress leads to placental insufficiency and the inability to cope with the demands of the growing fetus, eventually causing fetal death [34].

Hypoxia, hyperoxia, ischemia, and inflammation produce free radicals. In conditions of metabolic stress, such as ischemia/reperfusion, the percentage of oxygen that is reduced incompletely may increase, thus favoring the production of ROS by the mitochondria [35].

The placenta plays a crucial role in causing oxidative stress during pregnancy. It contains high levels of polyunsaturated fatty acids and is a plentiful source of peroxylipids, which are secreted into the maternal bloodstream. The placenta contains several antioxidant enzymes, such as superoxide glutathione peroxidase (GPx), catalase (CAT), glucose-6-phosphate dehydrogenase (G6PDH), and dismutase (SOD). On the other hand, the placental production of peroxylipids progressively decreases during the normal continuation of pregnancy, mainly due to the high activity of SOD and CAT. During a normal pregnancy, the placenta’s antioxidant defenses can effectively manage lipid peroxidation. 

The exact mechanism of cell damage in newborns due to hypoxia/ischemia is not fully understood. It is likely caused by an excessive release of neurotransmitters, reactive oxygen and nitrogen species (ROS/RNS), and the initiation of lipid peroxidation, triggering a cascade of events. At the cellular level, hypoxia/ischemia triggers a series of biochemical reactions that shift metabolism from oxidative to anaerobic, leading to the accumulation of NADH, FADH, H+ ions, and lactic acid.

If the asphyxiated insult continues, the fetus will not be able to maintain circulation to vital organs, which can lead to damage to the heart and brain. This damage is thought to be caused by the post-ischemic production of free radicals, nitric oxide, and inflammatory cytokines such as IL-6 [24].

The study by Many et al. revealed intense immunoreactivity for nitrotyrosine in invasive cytotrophoblasts in placental biopsies and vascular endothelium in the floating villi obtained from women with pre-eclampsia. The presence of nitrotyrosine suggests damage caused by peroxynitrite, a powerful nitrosative agent. The discovery of nitrotyrosine residues in the cellular components of pre-eclamptic placentas may indicate the increased production of the superoxide anion radical, as it combines with nitric oxide to produce peroxynitrite [36].

The results of our study show an increase in markers of OS in SIUD cases; these data are important in forensic pathology applications in cases of alleged malpractice. When a fetal death occurs during the last part of pregnancy, it may lead to legal action against the doctors or the hospital. As per Italian law, the forensic pathologist and obstetrician must analyze the medical documentation, perform the autopsy on the fetus, and analyze the placenta to evaluate the doctors’ management of the pregnancy. Very important in cases of SIUD, obtaining scientific evidence for evaluating the cases using immunohistochemical markers is crucial. The study aims to identify markers of oxidative stress to support clinical data in cases of medical malpractice in obstetrics.

## 5. Conclusions

Our study focused on the profound impact of oxidative stress in SIUD cases. These findings are important as SIUD often strikes in individuals who appear healthy, leading to an outcome that could not have been foreseen. The analysis of placenta samples revealed a substantial increase in oxidative stress in SIUD cases. Key players in placental oxidative damage are NOX2, 8HOGD, and nitrotyrosine. These insights could prove invaluable to forensic pathologists dealing with malpractice cases in obstetrics and studying SIUD cases.

OS is a pivotal factor in a range of pregnancy complications. The overstimulation of ROS can lead to IUGR, miscarriage, and spontaneous abortion at any stage of pregnancy. Placental oxidative stress is a significant contributor to adverse pregnancy outcomes, influenced by various factors such as genetics, maternal history, and environmental conditions.

This paper delves into the potential implications of oxidative stress-induced placental dysfunction in pregnancies complicated by sudden intrauterine unexplained death syndrome. The study presents compelling evidence for the strong link between OS and fetal death. This finding is particularly crucial in cases of suspected obstetric malpractice, where an unexpected intrauterine fetal death lacking a natural cause of death necessitates a thorough autoptic and histological analysis to determine the correct actions of the physicians. The immunohistochemistry of oxidative markers of the placenta is increasingly vital for forensic purposes, particularly in identifying anomalies in pregnancy. In our study, NOX2, 8HOGD, and nitrotyrosine are key players in placental oxidative damage. We demonstrated increased oxidative damage in the placentas from unexplained fetal death, as evidenced by an increase in the production of 8-OHdG (a biomarker of DNA oxidation), NOX2, and nitrotyrosine in stillbirth placentas compared with the control group.

## Figures and Tables

**Figure 1 cells-13-01347-f001:**
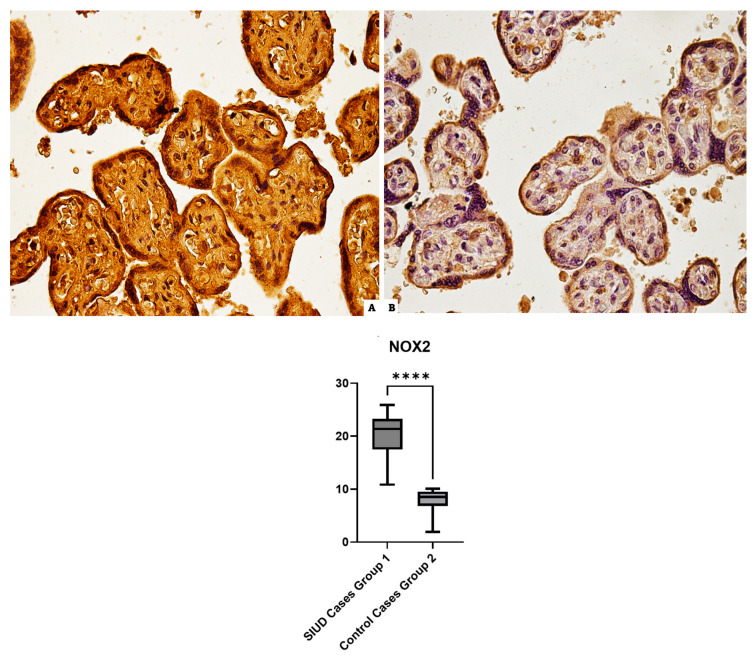
In the image, at 40× magnification, the immunohistochemical reaction against the anti-NOX2 antibody: (**A**) Group 1 (N = 15), the strong NOX2 diffuse immunopositivity localized in the placental tissue, central area. (**B**) Group 2 (N = 10), mild immunoreaction to NOX2 in the control tissue. In the graph at the bottom, the statistical representation, **** (statistically significant), *p* < 0.0001.

**Figure 2 cells-13-01347-f002:**
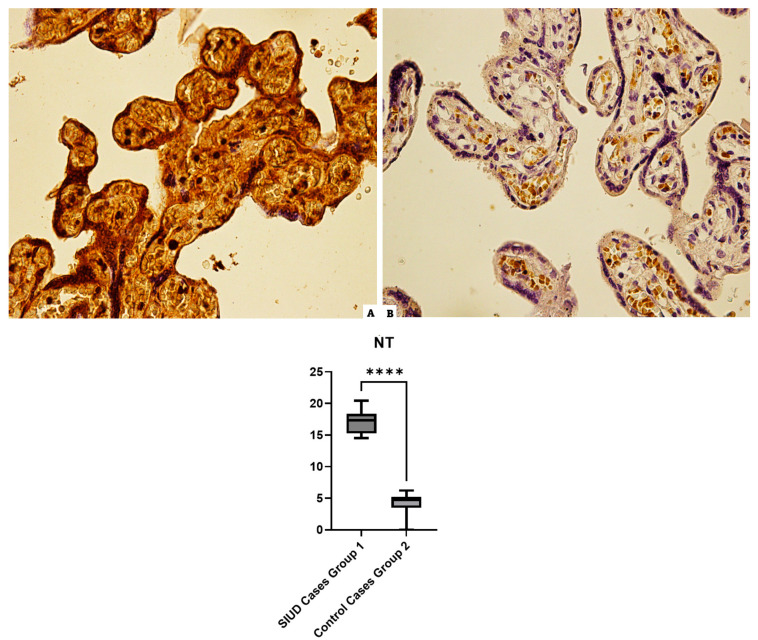
The results of the immunoreaction, at 40× magnification, to NT: (**A**) Group 1 (N = 15), overexpression of diffuse NT in the placenta of the cases, central part; (**B**) Group 2 (N = 10), low antibody reaction in the control tissue. In the lower part, the statistical comparison is represented graphically, **** (statistically significant): *p* < 0.0001.

**Figure 3 cells-13-01347-f003:**
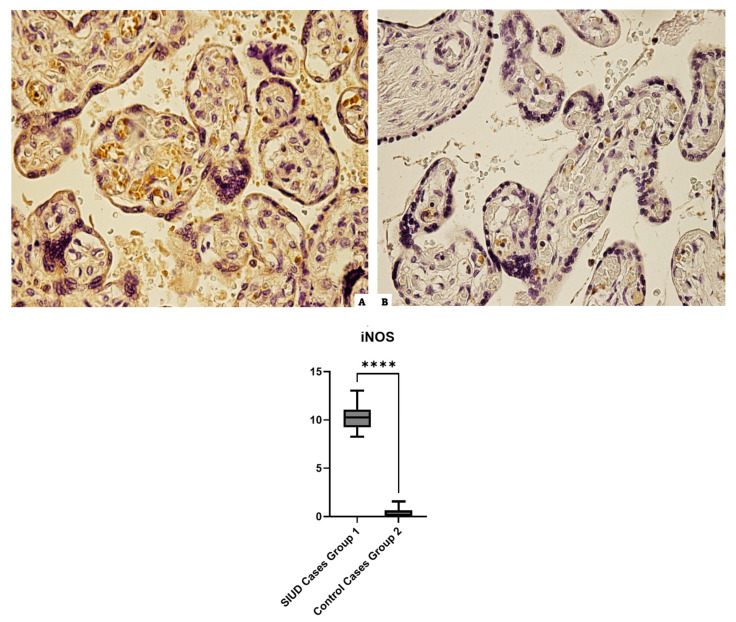
The results of immunohistochemical staining, at 40× magnification, with the marker iNOS: (**A**), in Group 1 (N = 15), the placental tissue expresses intermediate immunoreactivity for iNOS; (**B**) Group 2 (N = 10), minimal immunoreactivity in the control. The graphical representation of the statistical analysis is placed at the bottom of the figure, **** (statistically significant): *p* < 0.0001.

**Figure 4 cells-13-01347-f004:**
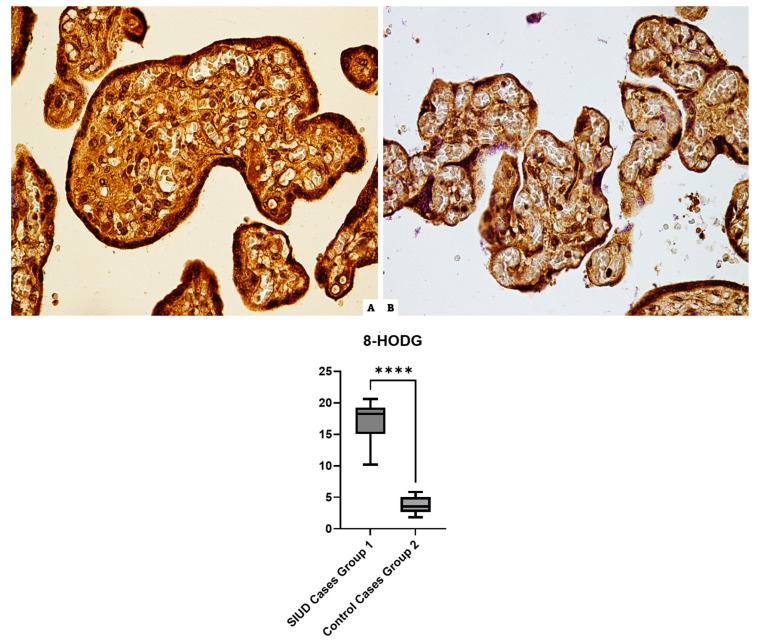
8-HODG immunohistochemical results in figure at 40× magnification: (**A**) Group 1 (N = 15): a diffuse and intense positive immunoreaction localized in placental tissue, central part, of 8-HODG; (**B**) Group 2 (N = 10): basal reaction in the control case. The graphical representation of the statistical analysis is collocated in the lower part of the figure, **** (statistically significant): *p* < 0.0001.

**Figure 5 cells-13-01347-f005:**
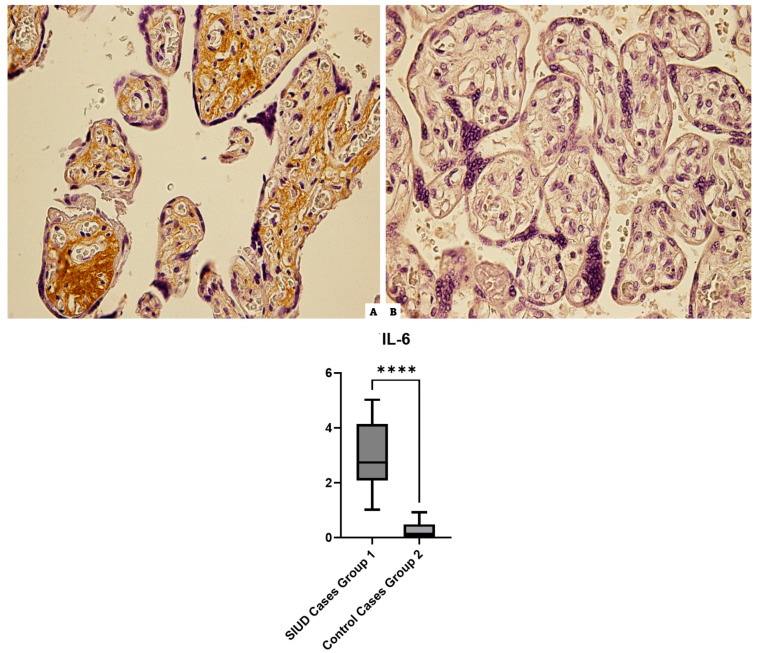
IL-6 immunohistochemical results at 40× magnification: (**A**) Group 1 (N = 15), shows a diffuse immunohistochemical intermediate reaction in the placental tissue of a case for the IL-6 marker; (**B**) in the image appreciates the very moderate reaction in Group 2 (N = 10), in the control case. The lower part of the figure is a graph of statistical analysis, **** (statistically significant): *p* < 0.0001.

**Figure 6 cells-13-01347-f006:**
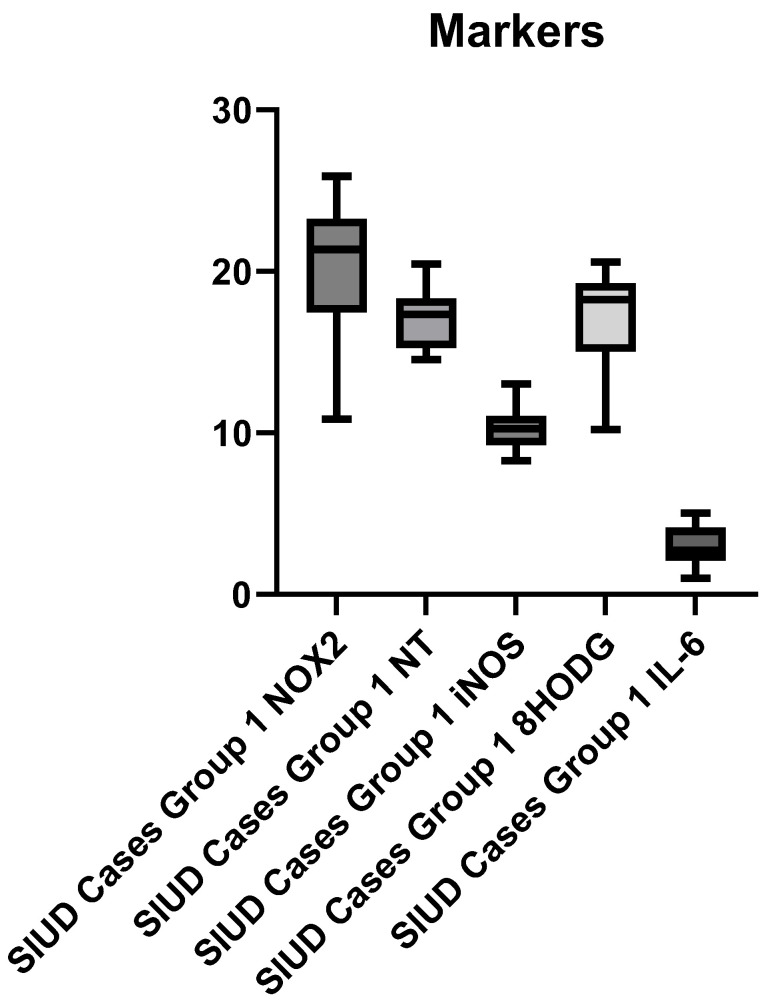
The graphical representation of the comparison of the expression between the various markers.

**Table 1 cells-13-01347-t001:** This table summarizes the salient information relating to the cases included in the study. For each case, the group in which it was included and the information relating to the pregnancy are reported.

Case Id	Group	Gender	Gestational Age (Weeks + Days)	Maternal Age (Years)	Type of Delivery	Weight (Grams)
1	1	Male	39	38	Vaginal	3296
2	1	Female	40	24	Vaginal	3200
3	1	Male	38 + 2	19	Cesarean	3580
4	1	Male	41	43	Cesarean	3000
5	1	Male	39 + 2	39	Vaginal	3450
6	1	Female	40	23	Vaginal	3200
7	1	Male	39 + 2	45	Cesarean	4000
8	1	Female	40	27	Vaginal	2900
9	1	Female	41 + 3	42	Vaginal	3120
10	1	Male	41 + 2	36	Cesarean	3369
11	1	Female	40	42	Cesarean	3560
12	1	Female	41	39	Cesarean	3600
13	1	Male	37 + 2	25	Vaginal	2950
14	1	Female	37 + 1	44	Vaginal	2890
15	1	Female	41 + 2	39	Cesarean	3050
16	2 Control	Male	40	30	Cesarean	3600
17	2 Control	Male	41	42	Vaginal	3450
18	2 Control	Female	40	24	Cesarean	3600
19	2 Control	Male	41	43	Vaginal	2920
20	2 Control	Female	39 + 3	25	Cesarean	3500
21	2 Control	Female	41 + 3	19	Vaginal	2960
22	2 Control	Female	41	45	Cesarean	3200
23	2 Control	Male	38 + 4	41	Vaginal	3600
24	2 Control	Female	37 + 4	39	Cesarean	2900
25	2 Control	Male	40 + 1	42	Vaginal	3820

## Data Availability

Data presented in the study are included in the article; further inquiries can be directed to the corresponding authors.

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
