# Peer review of "Sudden Intrauterine Unexplained Death (SIUD) and Oxidative Stress: Placental Immunohistochemical Markers"

_cells, 2024, doi:10.3390/cells13161347_

Round 1

Reviewer 1 Report

Comments and Suggestions for Authors

- Line 77: "Sudden intrauterine unexplained deaths" change to SIUD. After use the abbreviation for the first time, use always in the manuscript. 

- Line 82: IUGRs - the meaning is not in the text.

- Review of oxidative stress in the introduction is too extensive. I suggest to point out the main aspects of oxidative stress related to the work and the literature findings that support the objective of the paper.

- The introduction is too long with some paragraphs with only one sentence. I believe that it can be improved.

- Line 154: "All cases were delivered vaginally". Is that correct? The table 2 present some cesarean.

- How many sections were made per sample? There was any gap between the sections analyzed?

- There is no A and B in the images.

- What is the Y axis of the graphics?

- There is no *** in the graphic of the image 3.

- Add the n per group in the legends.

- How many slices were made per sample? 

 - Why the authors point out that some results are more proeminent? What do you mean with that? This difference is really important?

Reviewer 2 Report

Comments and Suggestions for Authors

Dear authors,

Thanks for the paper that is submitted that covers an interesting topic. I think this article attempts to test the markers of oxidative stress in cases of SIUD, considering that cases of unexplained intrauterine fetal death are often the subject of malpractice cases of great forensic interest.  I have major comments and edits that I would like to suggest to the authors. After using an anti-plagiarism tool, I have found, literally copied, some paragraphs and figures from the article published by this group: “Cardiac SARS-CoV-2 Infection, Involvement of Cytokines in Postmortem Immunohistochemical Study”.

The percentage of plagiarism for this article alone is 33%.

I attach the plagiarism report that shows this issue. Although the editors have already checked this item. 

Introduction:

Lines 59 and 77: Sudden intrauterine unexplained Death: SIDS has already been introduced in line 57

As an advice to authors, in scientific writing, one paragraph cannot be composed of a sentence. (lines 72-73 and lines 83-84, for example, there are more)

Table 1 should be introduced in the Methods section.

Line 154: “All cases were delivered vaginally” But in Table 2 the type of delivery includes cesarean section as a way of delivery.

The conclusion of this article can be controversial. Placenta samples analyzed after a SIDS can show high levels of oxidative stress as a consequence of death itself, not as a cause.

There are two authors with self-citations:

Neri: 2

Greco: 3

Round 2

Reviewer 2 Report

Comments and Suggestions for Authors

Dese authors,

Thanks again for providing a new version of

the article. I still think there is a bias in the

conclusion: The conclusion of this article can be controversial. Placenta samples analyzed after a SIDS can show high levels of oxidative stress as a consequence of death itself, not as a cause.

That is why I think this article should

be rejected.                                                           

Author Response

Comments 1 " I still think there is a bias in the conclusion: The conclusion of this article can be controversial. Placenta samples analyzed after a SIDS can show high levels of oxidative stress as a consequence of death itself, not as a cause. That is why I think this article should be rejected."

Response 1

After reading your comments, we critically analyzed the conclusion and widely changed the focus.

We revised our statement about the cause of death and better explained the sense of our study. We want to underline the importance of immunohistochemistry in forensic pathology not only for simple causes of death but as a tool to individuate markers useful for post-mortem diagnosis.

We work in Italy, where the post-mortem diagnosis in malpractice cases is frequently crucial. Therefore, the identification of markers is increasingly important in criminal and civil cases. The problem is increasingly felt in the obstetric field and requires studies and scientific support.

To avoid any potential misunderstanding, we would like to clarify that our study exclusively uses placenta samples from cases of Sudden Intrauterine Unexplained Death (SIUD). We do not use samples from cases of Sudden Infant Death Syndrome (SIDS). Our study focuses on two distinct groups: SIUD cases and Control cases.

Thank you for your clarification; we explain better in the conclusion; see the last version of the manuscript (conclusion section).

Round 3

Reviewer 2 Report

Comments and Suggestions for Authors

Dear Authors,

Thank you for addressing the review comments comprehensively. Looking forward to reading this article.